# Ophthalmic Manifestations among HIV Patients at the Main Tertiary Hospital in Freetown, Sierra Leone: A Cross-Sectional Study

Jalikatu Mustapha [1,2], Enanga Sonia Namanga [2], Baimba Idriss [1], Daniel Sesay [2], Darlinda F. Jiba [2], James B. W. Russell [1,2], Mathew J. Vandy [1,2], Gibrilla F. Deen [1,2], George A. Yendewa [3,4,5,†] and Sulaiman Lakoh [1,2,6,*,†]

1   College of Medicine and Allied Health Sciences, University of Sierra Leone, Freetown 00232, Sierra Leone; jalikam@yahoo.com (J.M.); baimbaridriss@gmail.com (B.I.); jamesbwrussell@gmail.com (J.B.W.R.); mathewjusuvandy@yahoo.co.uk (M.J.V.); gibrilladeen1960@yahoo.com (G.F.D.)
2   Ministry of Health and Sanitation, Government of Sierra Leone, Freetown 00232, Sierra Leone; enangasonia@yahoo.com (E.S.N.); danielsesay10@gmail.com (D.S.); darlindajiba.dj@gmail.com (D.F.J.)
3   Department of Medicine, Case Western Reserve University School of Medicine, Cleveland, OH 44106, USA; gay7@case.edu
4   Division of Infectious Diseases and HIV Medicine, University Hospitals Cleveland Medical Center, Cleveland, OH 44106, USA
5   Johns Hopkins Bloomberg School of Public Health, Baltimore, MD 21205, USA
6   Sustainable Health Systems Sierra Leone, Freetown 00232, Sierra Leone
*   Correspondence: lakoh2009@gmail.com
†   These authors contributed equally to this work.

**Abstract:** Ophthalmic diseases are common among people living with HIV (PLHIV) in developing countries. However, there are currently no published studies on ophthalmic complications among PLHIV in Sierra Leone. We conducted a cross-sectional study of PLHIV presenting at Connaught Hospital in Freetown, Sierra Leone from January through March 2020. Logistic regression was used to identify associations between ophthalmic manifestations and potential risk factors. A total of 103 PLHIV were studied (78.6% female, median age 41 years, 100% treatment-experienced). The median CD4 cell count was 374 cells/mm$^3$ and 76.7% were virologically suppressed. Overall, 44.7% of study participants had at least one ophthalmic complication and 51.5% had poor visual acuity in at least one eye. The most common conditions were dry eye (21.4%), cataract (20.4%), blepharitis (10.7%), nucleosclerosis (6.8%), conjunctivitis (5.8%), penguecula (5.8%), toxoplasmic retinochoroiditis (3.9%), and posterior vitreous detachment (2.9%). In multivariate logistic regression analysis, poor visual acuity (adjusted odds (aOR) 2.75, 95% confidence interval (CI) [1.12–6.78]; $p = 0.040$) and CD4 cell count < 100 cells/mm$^3$ (aOR 3.91, 95% CI [1.07–14.31]; $p = 0.028$) were independently associated with ophthalmic disease. A high proportion of PLHIV in this study had ophthalmic complications. This calls for greater integration of HIV and ophthalmologic care.

**Keywords:** ophthalmic lesions; ophthalmic manifestations; ophthalmic complications; ocular lesions; ocular manifestations; cataracts; Freetown; Sierra Leone

## 1. Introduction

With the advent of antiretroviral therapy (ART), HIV infection has become a chronic disease and people living with HIV (PLHIV) have a longer life expectancy [1]. The improvement in PLHIV survival has had unintended consequences, including an increased burden of ocular manifestations [2,3].

Ocular manifestations of HIV infection are common in low- and middle-income countries (LMICs). Studies from sub-Saharan Africa reported that 48% of PLHIV in Ghana [4]

and 70% of PLHIV in Tanzania [5] had ocular abnormalities. Elsewhere, ocular conditions have been reported in 53 to 65% of PLHIV at some point in their disease course [6,7].

The ocular complications seen in PLHIV may be related to the state of the immune system, as measured by the CD4 cell count [8]. More severe ocular disease may occur in advanced HIV [9]. For example, ocular toxoplasmosis occurs when the CD4 count falls below 200 cells/mm$^3$, and ocular cytomegalovirus occurs when the CD4 count falls below 100 cells/mm$^3$ [9]. Hence, these conditions can be treated effectively with a combination of specific therapy as well as a build-up of the immune system using highly active antiretroviral therapy (HAART) [10].

The ophthalmic complications of HIV infection can lead to visual impairment and blindness if untreated. A recent study has shown that up to 2% of PLHIV suffer from vision loss [11] and face challenges of visual impairment, including significant financial and psychosocial burdens [12]. Thus, screening for eye conditions should be part of the initial assessment of newly diagnosed HIV patients, as well as part of the routine follow-up for those living with the virus [10].

Sierra Leone has a national HIV prevalence of 1.7% [13], but, owing to programmatic challenges of HIV treatment in Sierra Leone, many patients are presenting late to the health facilities with advanced diseases and unintended consequences, including ocular complications [14,15]. The country does not have a system for routine assessment and monitoring of eye disease in PLHIV, mainly because of the lack of data to inform policies to improve eye care services for PLHIV in the country. A literature search conducted in February 2022 did not uncover any studies evaluating ophthalmic complications of PLHIV in Sierra Leone. This study will, for the first time, determine the pattern of ophthalmic manifestations of PLHIV in Sierra Leone, thereby providing the necessary evidence for training health care workers and initiating routine eye examinations.

The specific study objectives were to determine (a) the prevalence and types of ophthalmic abnormalities in PLHIV at Connaught Hospital and (b) the risk factors for ophthalmic abnormalities among PLHIV at Connaught hospital.

## 2. Materials and Methods

### 2.1. Study Design, Population, and Period

This is a cross-sectional descriptive study of PLHIV aged 18 years and above attending the HIV clinic at Connaught Hospital in Freetown between January and March 2020.

### 2.2. Study Settings

The study was conducted at the HIV and Eye Clinics of Connaught Hospital, University of Sierra Leone Teaching Hospital Complex in Freetown, Sierra Leone. Connaught Hospital is a public hospital that serves as the country's main referral facility and houses the only medical school in Sierra Leone. More than 5000 PLHIV are enrolled in ART, with approximately 10 new patients initiating ART per week.

### 2.3. Sampling and Recruitment

The sample size was determined using Fisher's formula ($n = z^2pq/d^2$). Here, '*n*' is the required minimum sample size; 'z' is the standard normal deviation, which is set conventionally at 1.96 and corresponds to a 95% confidence interval; 'p' is the estimated prevalence of ocular manifestation in PLHIV (4.0% based on what is reported in Nigeria) [16]; and 'd' is the maximum random sampling error acceptable, which was 0.05 precision at a 95% confidence interval. The calculated sample size '*n*' is 60. Adding a 10% for contingencies brought the minimum calculated sample size to 66, but 103 PLHIV were recruited.

The HIV clinic registration room was used to enumerate participants for the purposes of inclusion in the study. Consecutive PLHIV who came to the HIV Clinic at Connaught Hospital were informed of the study. Those who accepted to participate in the study were given more information and asked to sign a written consent form. PLHIV were asked

questions on demographic characteristics, HIV details, and ocular symptoms before being taken to the Eye Clinic for their ocular assessment.

A thorough eye assessment was conducted on all the consenting participants after the application of the tropicamide pupil dilator to aid proper visualization of the posterior part of the eye. We used Snellen's Visual Acuity chart to determine the visual acuity and a torch, loupes, and slit lamps to examine the ocular adnexal and anterior segments. The tear breakup time (TBUT) method was used to assess the ocular surface, and a TBUT of fewer than 10 s was considered indicative of dry eye. After pupil dilation with tropicamide drops, the posterior segment of the eye was examined using an indirect ophthalmoscope with a 20D lens and/or a slit lamp with a 90D lens, as appropriate.

### 2.4. Data Collection, Management, and Statistical Analyses

Sociodemographic and clinical data were collected using a standard questionnaire and medical records, entered into a Microsoft Excel spreadsheet, and stored in a password-protected computer. Statistical analyses were performed using the SPSS Version 28.0 (Armonk, NY; IBM Corp). For HIV-specific laboratory tests, we used the most recent CD4 cell count and HIV viral load collected within the last 6 months as current values. Additionally, we estimated missing test values using multiple imputation. Categorical variables were reported as frequencies (percentages) and associations assessed using Pearson's chi-square or Fisher's exact tests. Continuous variables were presented as medians (interquartile ranges, IQR) and associations assessed using the non-parametric independent samples Mann–Whitney U-test. Logistic regression was used to identify associations between ocular manifestations and potential risk factors. Variables that attained a *p*-value of <0.2 in the univariate analysis were included in the multivariate regression model. Associations were reported as crude (OR) and adjusted odds ratios (aOR) with 95% confidence intervals (CI), with statistical significance set at $p < 0.05$.

### 3. Results

#### 3.1. Socio-Demographic and Clinical Profile of the Study Participants

A total of 103 PLHIV were enrolled in the study. Table 1 displays the baseline characteristics of the study participants. The median age was 41 years (IQR 32–52), with most study participants being female (78.6%, 81/103), single (46.6%, 48/103), and employed in the informal sector (73.8%, 76/103). The median CD4 count was 374 cells/mm$^3$ (IQR 190–580) and the vast majority were virologically suppressed (76.7%, 79/103), with a substantial proportion of participants (39.8%, 41/103) having a history of opportunistic infections (OIs). Tenofovir-based regimens were the most commonly used ART, with a median ART duration of 5 years (IQR 1.7–9).

**Table 1.** Socio-demographic and clinical features of study participants, N = 103.

| Characteristics | N (%) |
|:---:|:---:|
| Age, years | |
| Median (IQR) | 41 (32–52) |
| 20–29 | 18 (17.5) |
| 30–39 | 27 (26.2) |
| 40–49 | 20 (19.4) |
| 50–59 | 31 (30.1) |
| ≥60 | 7 (6.8) |
| Gender | |
| Male | 22 (21.4) |
| Female | 81 (78.6) |

**Table 1.** *Cont.*

| Characteristics | N (%) |
|---|---|
| **Relationship status** | |
| Single | 48 (46.6) |
| Married | 39 (37.9) |
| Widowed | 16 (15.5) |
| **Employment status** | |
| Unemployed | 17 (16.5) |
| Informal sector | 76 (73.8) |
| Formal sector | 10 (9.7) |
| **Opportunistic infections *** | |
| Tuberculosis | 35 (34.0) |
| Kaposi sarcoma | 4 (3.9) |
| Herpes zoster | 2 (1.9) |
| ART regimens | |
| Tenofovir-based | 54 (52.4) |
| Zidovudine-based | 49 (47.6) |
| **ART duration, years** | |
| Median (IQR) | 5 (1.7–9) |
| <2 | 28 (27.2) |
| 2–4 | 19 (18.4) |
| 4–6 | 8 (7.8) |
| 6–8 | 10 (9.7) |
| >8 | 38 (36.9) |
| **CD4 count, cells/mm$^3$** | |
| Median (IQR) | 374 (190–580) |
| <100 | 16 (15.5) |
| 100–199 | 30 (29.1) |
| 200–349 | 14 (13.6) |
| ≥350 | 43 (41.8) |
| **Viral load, copies/mL** | |
| <1000 | 79 (76.7) |
| ≥1000 | 24 (23.3) |

* Not mutually exclusive, and not all patients had an opportunistic infection.

### 3.2. Presenting Complaints and Visual Acuity Findings

The presenting complaints or symptoms and visual acuity assessment findings are presented in Table 2. The most common symptoms (not mutually exclusive) were itching (28.2%, 29/103), poor vision (24.3%, 25/103), and photophobia (10.7%, 11/103), followed by foreign body sensation (7.8%, 8/103), eye pain (7.8%, 8/103), and red eye (6.8%, 7/103). Visual acuity was abnormal in 30.1% (31/103) of right eye and 37.9% (39/103) of left eye examinations. Additionally, 23.3% (24/103) and over half (51.5%, 53/103) had abnormal visual acuity in bilateral and at least one eye, respectively.

**Table 2.** Presenting symptoms and visual acuity assessments.

| Characteristics | N (%) |
|---|---|
| **Symptoms at presentation** **(Not mutually exclusive)** | |
| Itching | 29 (28.2) |
| Poor vision | 25 (24.3) |
| Photophobia | 11 (10.7) |

**Table 2.** *Cont.*

| Characteristics | N (%) |
|---|---|
| Foreign body sensation | 8 (7.8) |
| Eye pain | 8 (7.8) |
| Red eye | 7 (6.8) |
| Tearing | 4 (3.9) |
| Discharge | 4 (3.9) |
| Burning sensation | 2 (1.9) |
| Swelling | 2 (1.9) |
| Floaters | 1 (1.0) |
| Difficulty reading | 1 (1.0) |
| Abnormal visual acuity findings | |
| Right eye | 31 (30.1) |
| Left eye | 39 (37.9) |
| Bilateral eyes | 24 (23.3) |

### 3.3. Prevalence of Ophthalmic Manifestations

Table 3 summarizes the prevalence of ocular manifestations (not mutually exclusive) based on adnexal, anterior segment, and posterior segment findings. Overall, 44.7% (46/103) of study participants had at least one ocular diagnoses. In terms of individual findings, adnexal and anterior segment diseases were the most common diagnoses, with keratoconjunctivitis sicca or dry eye disease (21.4%, 22/103) and cataract (22.3%, 23/103) occurring in equal proportion. Other common adnexal and anterior segment findings included blepharitis (10.7%, 11/103), conjunctivitis (5.8%, 6/103), and penguecula (5.8%, 6/103). In comparison, posterior segment disease was relatively rare, with posterior vitreous detachment (7.8%, 8/103) and toxoplasmic retinochoroiditis (3.9%, 4/103) being the most common findings.

**Table 3.** Prevalence of ophthalmic manifestations.

| Diagnoses | N (%) |
|---|---|
| Overall (at least 1 ocular manifestation) | 46 (44.7) |
| Adnexal disease | |
| Dry eye | 22 (21.4) |
| Blepharitis | 11 (10.7) |
| Hyperemia | 1 (1.0) |
| Mucoid discharge | 1 (1.0) |
| Entropion/hypoglobus | 1 (1.0) |
| Macropapule | 1 (1.0) |
| Anterior segment disease | |
| Cataract | 23 (22.3) |
| Conjunctivitis | 6 (5.8) |
| Penguecula | 6 (5.8) |
| Pterygium | 3 (2.9) |
| Superficial punctate keratopathy | 2 (1.9) |
| Microvasculopathy | 2 (1.9) |
| Conjunctiva nevus | 1 (1.0) |
| Corneal epitheliopathy | 1 (1.0) |
| Posterior segment disease | |
| Posterior vitreous detachment | 8 (7.8) |
| Toxoplasmic retinochoroiditis | 4 (3.9) |
| Presbyopia | 2 (1.9) |
| HIV retinopathy | 2 (1.9) |
| Cotton wool spots | 2 (1.9) |
| Vitreous opacities | 1 (1.0) |

### 3.4. Factors Associated with Ophthalmic Manifestations

Factors associated with ophthalmic diagnoses are presented in Table 4. In multivariate logistic regression analysis, having abnormal visual acuity in at least one eye (aOR 2.75, 95% CI [1.12–6.78]; $p$ = 0.040) and CD4 cell count < 100 (aOR 3.91, 95% CI [1.07–14.31]; $p$ = 0.028) were independently associated with ophthalmic disease. There was a trend towards higher prevalence of ocular disease with age, which did not attain statistical significance ($p$ = 0.080).

**Table 4.** Risk factor assessment of ophthalmic manifestations.

| Risk Factors | Ocular Manifestations | | Univariate Analysis | | Multivariate Analysis | |
|---|---|---|---|---|---|---|
| | Yes | No | Crude Odds Ratio (95% CI) | *p*-Value | Adjusted Odds Ratio (95% CI) | *p*-Value |
| **Gender** | | | | | | |
| Male | 12 (22.6) | 10 (20.0) | 1.17 (0.46–3.01) | 0.744 | | |
| Female | 41 (77.4) | 40 (80.0) | Reference | | | |
| **Age > 55 years** | | | | | | |
| Yes | 13 (24.5) | 2 (4.0) | 7.80 (1.66–36.63) | 0.004 | 4.33 (0.84–22.35) | 0.080 |
| No | 40 (75.5) | 48 (96.0) | Reference | | Reference | |
| **Relationship status** | | | | | | |
| Single | 27 (50.9) | 21 (42.0) | 1.29 (0.41–4.00) | 0.664 | | |
| Married | 18 (34.0) | 21 (42.0) | 0.86 (0.27–2.75) | 0.795 | | |
| Widowed | 8 (15.1) | 8 (16.0) | Reference | | | |
| **Employment status** | | | | | | |
| Unemployed | 9 (17.0) | 8 (16.0) | 0.75 (0.15–3.65) | 0.722 | | |
| Informal sector | 38 (71.7) | 38 (76.0) | 0.67 (0.17–2.55) | 0.554 | | |
| Formal sector | 6 (11.3) | 4 (8.0) | Reference | | | |
| **Opportunistic infections** | | | | | | |
| Yes | 24 (45.3) | 17 (34.0) | 1.61 (0.72–3.56) | 0.242 | | |
| No | 29 (54.7) | 33 (66.0) | Reference | | | |
| **ART regimen** | | | | | | |
| Tenofovir-based | 25 (47.2) | 29 (58.0) | 0.65 (0.30–1.411) | 0.271 | | |
| Zidovudine-based | 28 (52.8) | 21 (42.0) | Reference | | | |
| **ART duration** | | | | | | |
| <2 years | 15 (28.3) | 13 (26.0) | 1.12 (0.47–2.68) | 0.793 | | |
| ≥2 years | 38 (71.7) | 37 (74.0) | Reference | | | |
| **Abnormal visual acuity in at least one eye** | | | | | | |
| Yes | 31 (58.5) | 15 (30.0) | 3.29 (1.46–7.43) | 0.004 | 2.75 (1.12–6.78) | 0.040 |
| No | 22 (41.5) | 35 (70.0) | Reference | | Reference | |
| **CD4 < 100 cells/mm$^3$** | | | | | | |
| Yes | 12 (22.6) | 4 (8.0) | 3.37 (1.01–11.26) | 0.040 | 3.91 (1.07–14.31) | 0.028 |
| No | 41 (77.4) | 46 (92.0) | Reference | | Reference | |
| **Viral load, copies/mL** | | | | | | |
| <1000 | 41 (77.4) | 38 (76.0) | 1.08 (0.43–2.69) | 0.871 | | |
| ≥1000 | 12 (22.6) | 12 (24.0) | Reference | | | |

## 4. Discussion

The first study on ophthalmic manifestations showed that 44.7% of PLHIV receiving care in the largest HIV unit in Sierra Leone had ocular manifestations, higher than reported from Ghana, Ethiopia, and India [17–19]. While the patients included in this study are ART-experienced, the high prevalence of eye disease reflects the existing health challenges of PLHIV in Sierra Leone, whether or not on antiretroviral therapy. Therefore, the findings urgently call for the integration of eye care into HIV services.

Similar to studies in Ethiopia and elsewhere in India, anterior segment lesions are the most common lesions reported in this study [8,20]. This finding provides a window of opportunity for early intervention to detect more sinister lesions because anterior segment lesions are less likely to cause blindness. Among the diseases of the anterior segment, keratoconjunctivitis sicca or dry eye is the most common. Dry eye is sometimes a manifestation of a dysregulated immune response due to an autoimmune exocrine disease, including Sjögren's syndrome [20,21]. Historically, HIV has been associated with Sjögren's syndrome, which is a major cause of dry eye disease [22]. In an earlier study, HIV infection was shown to cause a clinical syndrome similar to Sjögren's syndrome, but without the formation of classic autoantibodies [23]. Whether the dry eyes reported in this patient population are immune-mediated is a question that needs to be answered in future research. In this context, however, it is worth paying attention to autoimmune diseases in HIV patients. Posterior segment lesions, though they are relatively uncommon in this study, unlike studies in Turkey and India, are significant, considering their high tendency to cause blindness [24,25].

Itching, poor vision, photophobia, foreign body sensation, eye pain, and red eye are common symptoms of eye disease in our study population. Similar findings have been reported in previous studies of ocular abnormalities in PLHIV [10,11]. Currently, the HIV treatment case file, which documents the clinical details of HIV patients, does not include a detailed eye assessment. Therefore, it is worthwhile to establish structures for routine ocular assessment and for appropriate and timely management of patients with ocular lesions [26].

The National AIDS Control Program conceived the development of the Consolidated Guidelines on HIV Prevention, Diagnosis, Treatment, and Care in Sierra Leone to improve on the care of PLHIV in the country, but did not mention the management of ophthalmic manifestations in this population [26]. This may be explained by the fact that the HIV program in Sierra Leone operates within a limited scope of indicators that, in most cases, does not expand beyond the needs of the Global Fund. Advocacy is needed to ensure that dedicated resources are allocated to the implementation of other health issues of PLHIV in the country.

A CD4 cell count of less than 100 is a strong predictor of ophthalmic diagnosis, similar to previous studies [10]. The main challenge of this finding is the high burden of advanced disease and late diagnosis in this setting [14,15]. The way forward would be to prioritize interventions that will prevent late-stage presentation and advanced HIV disease.

Unlike reports in other studies, opportunistic ocular infections were rarely reported in our study [27]. This is expected because, as previously reported, many of the patients in this study were virologically suppressed, and their ocular abnormalities may be explained by a combination of factors, including the toxicity of antiretroviral therapy and the drugs used to treat OIs [28].

Although this study provides the first evidence of ocular abnormalities in PLHIV in Sierra Leone and highlights opportunities for public health and clinical intervention, it has some limitations. Our study did not assess ocular abnormalities in ART-naïve patients, which may underestimate its true burden. The study also excluded the ART-experienced population with comorbidities; thus, we have not been able to understand their contribution to the burden of ophthalmic lesions in PLHIV. Likewise, using a relatively small sample size to conduct this study in a single setting makes it impossible to generalize the findings to people living with HIV in Sierra Leone.

## 5. Conclusions

A high proportion of PLHIV (44.6%) had ophthalmic complications in Sierra Leone, despite being treatment-experienced on appropriate ART and the majority being virologically suppressed. Adnexal and anterior segment diseases were the most common diagnoses, with keratoconjunctivitis sicca, i.e., dry eye disease, and cataract occurring in about 1 in 4 PLHIV. Other common adnexal and posterior segment findings (<10%) included blepharitis, conjunctivitis, and penguecula. Posterior segment disease was relatively rare (<4%), with toxoplasmic retinochoroiditis and posterior vitreous detachment being the most common. The most important determinants of ophthalmic disease were poor visual acuity and severe immunosuppression (CD4 count < 100 cells/mm$^3$), which were expected findings. This calls for greater integration of HIV and ophthalmologic care to achieve better clinical outcomes and improved quality of life for PLHIV in the country.

**Author Contributions:** Conceptualization: S.L., J.M., E.S.N., M.J.V. and B.I. Methodology: G.A.Y., S.L., J.M., B.I., D.F.J. and E.S.N. Formal analysis: G.A.Y. and D.F.J. Data curation: E.S.N., J.M., B.I., M.J.V., D.S. and S.L. Supervision: J.B.W.R., G.F.D. and M.J.V. Writing original draft preparation: S.L., G.A.Y. and J.M. Writing—review and editing: M.J.V., B.I., J.B.W.R., G.F.D. and D.S. All authors have read and agreed to the published version of the manuscript.

**Funding:** This research received no external funding.

**Institutional Review Board Statement:** Ethical approval was obtained from the Sierra Leone Ethics and Scientific Review Committee of the Ministry of Health and Sanitation, Government of Sierra Leone. Written informed consent was not required for this retrospective study as it has been waived by the Sierra Leone Ethics and Scientific Review Committee of the Ministry of Health and Sanitation.

**Informed Consent Statement:** Not applicable.

**Data Availability Statement:** The data supporting this article are available in the repository of University of Sierra Leone and will be made available on request to the corresponding author when required.

**Acknowledgments:** We acknowledge the cooperation of the staff at the HIV and Eye clinics, patients, and patients' relatives at the hospitals where the study was conducted.

**Conflicts of Interest:** G.A.Y. reports salary support from the National Institutes of Health/AIDS Clinical Trials Group under Award Numbers 5UM1AI068636-15, 5UM1AI069501-09, and AI068636(150GYD212), and consultancy fees from Pfizer. Other authors have no conflict of interest related to this study.

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
