# Peer review of "Ophthalmic Manifestations among HIV Patients at the Main Tertiary Hospital in Freetown, Sierra Leone: A Cross-Sectional Study"

_venereology, doi:10.3390/venereology1020011_

Round 1

Reviewer 1 Report

The authors conducted a cross-sectional study to determine 1) the prevalence and types of ophthalmic abnormalities in PLHIV, 2) the risk factors for ophthalmic abnormalities among PLHIV at the HIV and eye clinics of Connaught Hospital, University of Sierra Leone Teaching Hospital complex in Freetown, Sierra Leone. It will be helpful that the authors can clarify whether the study participants can represent the PLHIV at the Connaught Hospital and the findings can be linked or generalized to the PLHIV in the country.

Line 73-81, The authors mentioned that this is the first study in Sierra Leone to determine the pattern of ophthalmic manifestations of PLHIV and to provide the necessary evidence to the policy makers for training health care workers and initiating routine eye examinations. Is the Connaught Hospital, University of Sierra Leone Teach Hospital complex chosen for the study because of HIV prevalence in Freetown or/and have the largest number of PLHIV patients in the country? What is the percentage of PLHIV in Sierra Leone treated in this hospital? Can the results found among PLHIV at the Connaught Hospital be generalized to other hospital in Sierra Leone?

Line 94, the authors mentioned that more than 5000 PLHIV are enrolled in ART, with approximately 10 new patients initiating ART per week in the Connaught Hospital. However, there were only 103 PLHIV were enrolled in the study.  Was there a target sample size?  What might be the reasons that there are only 103 PLHIV (2%) enrolled in the study?

Line 134, Were the distributions of socio-demographic and clinical features of these 103 study participants similar to the other PLHIV who did not participate in this study? I am wondering if the finding from these 103 participants be generalized to the other 4900 PLHIV who did not participate in this study?

Line 161, Please consider using “proportion” to replace “prevalence”. Based on the section 2.3, the study participants are convenient samples. It is likely that the PLHIV with eye problem are more likely to participate in this study (selection bias). There is also no information to know if the 103 PLHIV can represent the other 4900 PLHIV at this hospital.

Table 1, Please check the precents (%). Some of the categories do not sum up to 1.  For example, unemployment status, opportunistic infections, etc.

Table 4, the second column, consider presenting raw percentages which align better with univariate and multivariable analysis.

The authors conducted a cross-sectional study to determine 1) the prevalence and types of ophthalmic abnormalities in PLHIV, 2) the risk factors for ophthalmic abnormalities among PLHIV at the HIV and eye clinics of Connaught Hospital, University of Sierra Leone Teaching Hospital complex in Freetown, Sierra Leone. It will be helpful that the authors can clarify whether the study participants can represent the PLHIV at the Connaught Hospital and the findings can be linked or generalized to the PLHIV in the country.

Line 73-81, The authors mentioned that this is the first study in Sierra Leone to determine the pattern of ophthalmic manifestations of PLHIV and to provide the necessary evidence to the policy makers for training health care workers and initiating routine eye examinations. Is the Connaught Hospital, University of Sierra Leone Teach Hospital complex chosen for the study because of HIV prevalence in Freetown or/and have the largest number of PLHIV patients in the country? What is the percentage of PLHIV in Sierra Leone treated in this hospital? Can the results found among PLHIV at the Connaught Hospital be generalized to other hospital in Sierra Leone?

Line 94, the authors mentioned that more than 5000 PLHIV are enrolled in ART, with approximately 10 new patients initiating ART per week in the Connaught Hospital. However, there were only 103 PLHIV were enrolled in the study.  Was there a target sample size?  What might be the reasons that there are only 103 PLHIV (2%) enrolled in the study?

Line 134, Were the distributions of socio-demographic and clinical features of these 103 study participants similar to the other PLHIV who did not participate in this study? I am wondering if the finding from these 103 participants be generalized to the other 4900 PLHIV who did not participate in this study?

Line 161, Please consider using “proportion” to replace “prevalence”. Based on the section 2.3, the study participants are convenient samples. It is likely that the PLHIV with eye problem are more likely to participate in this study (selection bias). There is also no information to know if the 103 PLHIV can represent the other 4900 PLHIV at this hospital.

Table 1, Please check the precents (%). Some of the categories do not sum up to 1.  For example, unemployment status, opportunistic infections, etc.

Table 4, the second column, consider presenting raw percentages which align better with univariate and multivariable analysis.

The authors conducted a cross-sectional study to determine 1) the prevalence and types of ophthalmic abnormalities in PLHIV, 2) the risk factors for ophthalmic abnormalities among PLHIV at the HIV and eye clinics of Connaught Hospital, University of Sierra Leone Teaching Hospital complex in Freetown, Sierra Leone. It will be helpful that the authors can clarify whether the study participants can represent the PLHIV at the Connaught Hospital and the findings can be linked or generalized to the PLHIV in the country.

Line 73-81, The authors mentioned that this is the first study in Sierra Leone to determine the pattern of ophthalmic manifestations of PLHIV and to provide the necessary evidence to the policy makers for training health care workers and initiating routine eye examinations. Is the Connaught Hospital, University of Sierra Leone Teach Hospital complex chosen for the study because of HIV prevalence in Freetown or/and have the largest number of PLHIV patients in the country? What is the percentage of PLHIV in Sierra Leone treated in this hospital? Can the results found among PLHIV at the Connaught Hospital be generalized to other hospital in Sierra Leone?

Line 94, the authors mentioned that more than 5000 PLHIV are enrolled in ART, with approximately 10 new patients initiating ART per week in the Connaught Hospital. However, there were only 103 PLHIV were enrolled in the study.  Was there a target sample size?  What might be the reasons that there are only 103 PLHIV (2%) enrolled in the study?

Line 134, Were the distributions of socio-demographic and clinical features of these 103 study participants similar to the other PLHIV who did not participate in this study? I am wondering if the finding from these 103 participants be generalized to the other 4900 PLHIV who did not participate in this study?

Line 161, Please consider using “proportion” to replace “prevalence”. Based on the section 2.3, the study participants are convenient samples. It is likely that the PLHIV with eye problem are more likely to participate in this study (selection bias). There is also no information to know if the 103 PLHIV can represent the other 4900 PLHIV at this hospital.

Table 1, Please check the precents (%). Some of the categories do not sum up to 1.  For example, unemployment status, opportunistic infections, etc.

Table 4, the second column, consider presenting raw percentages which align better with univariate and multivariable analysis.

The authors conducted a cross-sectional study to determine 1) the prevalence and types of ophthalmic abnormalities in PLHIV, 2) the risk factors for ophthalmic abnormalities among PLHIV at the HIV and eye clinics of Connaught Hospital, University of Sierra Leone Teaching Hospital complex in Freetown, Sierra Leone. It will be helpful that the authors can clarify whether the study participants can represent the PLHIV at the Connaught Hospital and the findings can be linked or generalized to the PLHIV in the country.

Line 73-81, The authors mentioned that this is the first study in Sierra Leone to determine the pattern of ophthalmic manifestations of PLHIV and to provide the necessary evidence to the policy makers for training health care workers and initiating routine eye examinations. Is the Connaught Hospital, University of Sierra Leone Teach Hospital complex chosen for the study because of HIV prevalence in Freetown or/and have the largest number of PLHIV patients in the country? What is the percentage of PLHIV in Sierra Leone treated in this hospital? Can the results found among PLHIV at the Connaught Hospital be generalized to other hospital in Sierra Leone?

Line 94, the authors mentioned that more than 5000 PLHIV are enrolled in ART, with approximately 10 new patients initiating ART per week in the Connaught Hospital. However, there were only 103 PLHIV were enrolled in the study.  Was there a target sample size?  What might be the reasons that there are only 103 PLHIV (2%) enrolled in the study?

Line 134, Were the distributions of socio-demographic and clinical features of these 103 study participants similar to the other PLHIV who did not participate in this study? I am wondering if the finding from these 103 participants be generalized to the other 4900 PLHIV who did not participate in this study?

Line 161, Please consider using “proportion” to replace “prevalence”. Based on the section 2.3, the study participants are convenient samples. It is likely that the PLHIV with eye problem are more likely to participate in this study (selection bias). There is also no information to know if the 103 PLHIV can represent the other 4900 PLHIV at this hospital.

Table 1, Please check the precents (%). Some of the categories do not sum up to 1.  For example, unemployment status, opportunistic infections, etc.

Table 4, the second column, consider presenting raw percentages which align better with univariate and multivariable analysis.

The authors conducted a cross-sectional study to determine 1) the prevalence and types of ophthalmic abnormalities in PLHIV, 2) the risk factors for ophthalmic abnormalities among PLHIV at the HIV and eye clinics of Connaught Hospital, University of Sierra Leone Teaching Hospital complex in Freetown, Sierra Leone. It will be helpful that the authors can clarify whether the study participants can represent the PLHIV at the Connaught Hospital and the findings can be linked or generalized to the PLHIV in the country.

Line 73-81, The authors mentioned that this is the first study in Sierra Leone to determine the pattern of ophthalmic manifestations of PLHIV and to provide the necessary evidence to the policy makers for training health care workers and initiating routine eye examinations. Is the Connaught Hospital, University of Sierra Leone Teach Hospital complex chosen for the study because of HIV prevalence in Freetown or/and have the largest number of PLHIV patients in the country? What is the percentage of PLHIV in Sierra Leone treated in this hospital? Can the results found among PLHIV at the Connaught Hospital be generalized to other hospital in Sierra Leone?

Line 94, the authors mentioned that more than 5000 PLHIV are enrolled in ART, with approximately 10 new patients initiating ART per week in the Connaught Hospital. However, there were only 103 PLHIV were enrolled in the study.  Was there a target sample size?  What might be the reasons that there are only 103 PLHIV (2%) enrolled in the study?

Line 134, Were the distributions of socio-demographic and clinical features of these 103 study participants similar to the other PLHIV who did not participate in this study? I am wondering if the finding from these 103 participants be generalized to the other 4900 PLHIV who did not participate in this study?

Line 161, Please consider using “proportion” to replace “prevalence”. Based on the section 2.3, the study participants are convenient samples. It is likely that the PLHIV with eye problem are more likely to participate in this study (selection bias). There is also no information to know if the 103 PLHIV can represent the other 4900 PLHIV at this hospital.

Table 1, Please check the precents (%). Some of the categories do not sum up to 1.  For example, unemployment status, opportunistic infections, etc.

Table 4, the second column, consider presenting raw percentages which align better with univariate and multivariable analysis.

The authors conducted a cross-sectional study to determine 1) the prevalence and types of ophthalmic abnormalities in PLHIV, 2) the risk factors for ophthalmic abnormalities among PLHIV at the HIV and eye clinics of Connaught Hospital, University of Sierra Leone Teaching Hospital complex in Freetown, Sierra Leone. It will be helpful that the authors can clarify whether the study participants can represent the PLHIV at the Connaught Hospital and the findings can be linked or generalized to the PLHIV in the country.

Line 73-81, The authors mentioned that this is the first study in Sierra Leone to determine the pattern of ophthalmic manifestations of PLHIV and to provide the necessary evidence to the policy makers for training health care workers and initiating routine eye examinations. Is the Connaught Hospital, University of Sierra Leone Teach Hospital complex chosen for the study because of HIV prevalence in Freetown or/and have the largest number of PLHIV patients in the country? What is the percentage of PLHIV in Sierra Leone treated in this hospital? Can the results found among PLHIV at the Connaught Hospital be generalized to other hospital in Sierra Leone?

Line 94, the authors mentioned that more than 5000 PLHIV are enrolled in ART, with approximately 10 new patients initiating ART per week in the Connaught Hospital. However, there were only 103 PLHIV were enrolled in the study.  Was there a target sample size?  What might be the reasons that there are only 103 PLHIV (2%) enrolled in the study?

Line 134, Were the distributions of socio-demographic and clinical features of these 103 study participants similar to the other PLHIV who did not participate in this study? I am wondering if the finding from these 103 participants be generalized to the other 4900 PLHIV who did not participate in this study?

Line 161, Please consider using “proportion” to replace “prevalence”. Based on the section 2.3, the study participants are convenient samples. It is likely that the PLHIV with eye problem are more likely to participate in this study (selection bias). There is also no information to know if the 103 PLHIV can represent the other 4900 PLHIV at this hospital.

Table 1, Please check the precents (%). Some of the categories do not sum up to 1.  For example, unemployment status, opportunistic infections, etc.

Table 4, the second column, consider presenting raw percentages which align better with univariate and multivariable analysis.

The authors conducted a cross-sectional study to determine 1) the prevalence and types of ophthalmic abnormalities in PLHIV, 2) the risk factors for ophthalmic abnormalities among PLHIV at the HIV and eye clinics of Connaught Hospital, University of Sierra Leone Teaching Hospital complex in Freetown, Sierra Leone. It will be helpful that the authors can clarify whether the study participants can represent the PLHIV at the Connaught Hospital and the findings can be linked or generalized to the PLHIV in the country.

Line 73-81, The authors mentioned that this is the first study in Sierra Leone to determine the pattern of ophthalmic manifestations of PLHIV and to provide the necessary evidence to the policy makers for training health care workers and initiating routine eye examinations. Is the Connaught Hospital, University of Sierra Leone Teach Hospital complex chosen for the study because of HIV prevalence in Freetown or/and have the largest number of PLHIV patients in the country? What is the percentage of PLHIV in Sierra Leone treated in this hospital? Can the results found among PLHIV at the Connaught Hospital be generalized to other hospital in Sierra Leone?

Line 94, the authors mentioned that more than 5000 PLHIV are enrolled in ART, with approximately 10 new patients initiating ART per week in the Connaught Hospital. However, there were only 103 PLHIV were enrolled in the study.  Was there a target sample size?  What might be the reasons that there are only 103 PLHIV (2%) enrolled in the study?

Line 134, Were the distributions of socio-demographic and clinical features of these 103 study participants similar to the other PLHIV who did not participate in this study? I am wondering if the finding from these 103 participants be generalized to the other 4900 PLHIV who did not participate in this study?

Line 161, Please consider using “proportion” to replace “prevalence”. Based on the section 2.3, the study participants are convenient samples. It is likely that the PLHIV with eye problem are more likely to participate in this study (selection bias). There is also no information to know if the 103 PLHIV can represent the other 4900 PLHIV at this hospital.

Table 1, Please check the precents (%). Some of the categories do not sum up to 1.  For example, unemployment status, opportunistic infections, etc.

Table 4, the second column, consider presenting raw percentages which align better with univariate and multivariable analysis.

The authors conducted a cross-sectional study to determine 1) the prevalence and types of ophthalmic abnormalities in PLHIV, 2) the risk factors for ophthalmic abnormalities among PLHIV at the HIV and eye clinics of Connaught Hospital, University of Sierra Leone Teaching Hospital complex in Freetown, Sierra Leone. It will be helpful that the authors can clarify whether the study participants can represent the PLHIV at the Connaught Hospital and the findings can be linked or generalized to the PLHIV in the country.

Line 73-81, The authors mentioned that this is the first study in Sierra Leone to determine the pattern of ophthalmic manifestations of PLHIV and to provide the necessary evidence to the policy makers for training health care workers and initiating routine eye examinations. Is the Connaught Hospital, University of Sierra Leone Teach Hospital complex chosen for the study because of HIV prevalence in Freetown or/and have the largest number of PLHIV patients in the country? What is the percentage of PLHIV in Sierra Leone treated in this hospital? Can the results found among PLHIV at the Connaught Hospital be generalized to other hospital in Sierra Leone?

Line 94, the authors mentioned that more than 5000 PLHIV are enrolled in ART, with approximately 10 new patients initiating ART per week in the Connaught Hospital. However, there were only 103 PLHIV were enrolled in the study.  Was there a target sample size?  What might be the reasons that there are only 103 PLHIV (2%) enrolled in the study?

Line 134, Were the distributions of socio-demographic and clinical features of these 103 study participants similar to the other PLHIV who did not participate in this study? I am wondering if the finding from these 103 participants be generalized to the other 4900 PLHIV who did not participate in this study?

Line 161, Please consider using “proportion” to replace “prevalence”. Based on the section 2.3, the study participants are convenient samples. It is likely that the PLHIV with eye problem are more likely to participate in this study (selection bias). There is also no information to know if the 103 PLHIV can represent the other 4900 PLHIV at this hospital.

Table 1, Please check the precents (%). Some of the categories do not sum up to 1.  For example, unemployment status, opportunistic infections, etc.

Table 4, the second column, consider presenting raw percentages which align better with univariate and multivariable analysis.

The authors conducted a cross-sectional study to determine 1) the prevalence and types of ophthalmic abnormalities in PLHIV, 2) the risk factors for ophthalmic abnormalities among PLHIV at the HIV and eye clinics of Connaught Hospital, University of Sierra Leone Teaching Hospital complex in Freetown, Sierra Leone. It will be helpful that the authors can clarify whether the study participants can represent the PLHIV at the Connaught Hospital and the findings can be linked or generalized to the PLHIV in the country.

Line 73-81, The authors mentioned that this is the first study in Sierra Leone to determine the pattern of ophthalmic manifestations of PLHIV and to provide the necessary evidence to the policy makers for training health care workers and initiating routine eye examinations. Is the Connaught Hospital, University of Sierra Leone Teach Hospital complex chosen for the study because of HIV prevalence in Freetown or/and have the largest number of PLHIV patients in the country? What is the percentage of PLHIV in Sierra Leone treated in this hospital? Can the results found among PLHIV at the Connaught Hospital be generalized to other hospital in Sierra Leone?

Line 94, the authors mentioned that more than 5000 PLHIV are enrolled in ART, with approximately 10 new patients initiating ART per week in the Connaught Hospital. However, there were only 103 PLHIV were enrolled in the study.  Was there a target sample size?  What might be the reasons that there are only 103 PLHIV (2%) enrolled in the study?

Line 134, Were the distributions of socio-demographic and clinical features of these 103 study participants similar to the other PLHIV who did not participate in this study? I am wondering if the finding from these 103 participants be generalized to the other 4900 PLHIV who did not participate in this study?

Line 161, Please consider using “proportion” to replace “prevalence”. Based on the section 2.3, the study participants are convenient samples. It is likely that the PLHIV with eye problem are more likely to participate in this study (selection bias). There is also no information to know if the 103 PLHIV can represent the other 4900 PLHIV at this hospital.

Table 1, Please check the precents (%). Some of the categories do not sum up to 1.  For example, unemployment status, opportunistic infections, etc.

Table 4, the second column, consider presenting raw percentages which align better with univariate and multivariable analysis.

The authors conducted a cross-sectional study to determine 1) the prevalence and types of ophthalmic abnormalities in PLHIV, 2) the risk factors for ophthalmic abnormalities among PLHIV at the HIV and eye clinics of Connaught Hospital, University of Sierra Leone Teaching Hospital complex in Freetown, Sierra Leone. It will be helpful that the authors can clarify whether the study participants can represent the PLHIV at the Connaught Hospital and the findings can be linked or generalized to the PLHIV in the country.

Line 73-81, The authors mentioned that this is the first study in Sierra Leone to determine the pattern of ophthalmic manifestations of PLHIV and to provide the necessary evidence to the policy makers for training health care workers and initiating routine eye examinations. Is the Connaught Hospital, University of Sierra Leone Teach Hospital complex chosen for the study because of HIV prevalence in Freetown or/and have the largest number of PLHIV patients in the country? What is the percentage of PLHIV in Sierra Leone treated in this hospital? Can the results found among PLHIV at the Connaught Hospital be generalized to other hospital in Sierra Leone?

Line 94, the authors mentioned that more than 5000 PLHIV are enrolled in ART, with approximately 10 new patients initiating ART per week in the Connaught Hospital. However, there were only 103 PLHIV were enrolled in the study.  Was there a target sample size?  What might be the reasons that there are only 103 PLHIV (2%) enrolled in the study?

Line 134, Were the distributions of socio-demographic and clinical features of these 103 study participants similar to the other PLHIV who did not participate in this study? I am wondering if the finding from these 103 participants be generalized to the other 4900 PLHIV who did not participate in this study?

Line 161, Please consider using “proportion” to replace “prevalence”. Based on the section 2.3, the study participants are convenient samples. It is likely that the PLHIV with eye problem are more likely to participate in this study (selection bias). There is also no information to know if the 103 PLHIV can represent the other 4900 PLHIV at this hospital.

Table 1, Please check the precents (%). Some of the categories do not sum up to 1.  For example, unemployment status, opportunistic infections, etc.

Table 4, the second column, consider presenting raw percentages which align better with univariate and multivariable analysis.

The authors conducted a cross-sectional study to determine 1) the prevalence and types of ophthalmic abnormalities in PLHIV, 2) the risk factors for ophthalmic abnormalities among PLHIV at the HIV and eye clinics of Connaught Hospital, University of Sierra Leone Teaching Hospital complex in Freetown, Sierra Leone. It will be helpful that the authors can clarify whether the study participants can represent the PLHIV at the Connaught Hospital and the findings can be linked or generalized to the PLHIV in the country.

Line 73-81, The authors mentioned that this is the first study in Sierra Leone to determine the pattern of ophthalmic manifestations of PLHIV and to provide the necessary evidence to the policy makers for training health care workers and initiating routine eye examinations. Is the Connaught Hospital, University of Sierra Leone Teach Hospital complex chosen for the study because of HIV prevalence in Freetown or/and have the largest number of PLHIV patients in the country? What is the percentage of PLHIV in Sierra Leone treated in this hospital? Can the results found among PLHIV at the Connaught Hospital be generalized to other hospital in Sierra Leone?

Line 94, the authors mentioned that more than 5000 PLHIV are enrolled in ART, with approximately 10 new patients initiating ART per week in the Connaught Hospital. However, there were only 103 PLHIV were enrolled in the study.  Was there a target sample size?  What might be the reasons that there are only 103 PLHIV (2%) enrolled in the study?

Line 134, Were the distributions of socio-demographic and clinical features of these 103 study participants similar to the other PLHIV who did not participate in this study? I am wondering if the finding from these 103 participants be generalized to the other 4900 PLHIV who did not participate in this study?

Line 161, Please consider using “proportion” to replace “prevalence”. Based on the section 2.3, the study participants are convenient samples. It is likely that the PLHIV with eye problem are more likely to participate in this study (selection bias). There is also no information to know if the 103 PLHIV can represent the other 4900 PLHIV at this hospital.

Table 1, Please check the precents (%). Some of the categories do not sum up to 1.  For example, unemployment status, opportunistic infections, etc.

Table 4, the second column, consider presenting raw percentages which align better with univariate and multivariable analysis.

The authors conducted a cross-sectional study to determine 1) the prevalence and types of ophthalmic abnormalities in PLHIV, 2) the risk factors for ophthalmic abnormalities among PLHIV at the HIV and eye clinics of Connaught Hospital, University of Sierra Leone Teaching Hospital complex in Freetown, Sierra Leone. It will be helpful that the authors can clarify whether the study participants can represent the PLHIV at the Connaught Hospital and the findings can be linked or generalized to the PLHIV in the country.

Line 73-81, The authors mentioned that this is the first study in Sierra Leone to determine the pattern of ophthalmic manifestations of PLHIV and to provide the necessary evidence to the policy makers for training health care workers and initiating routine eye examinations. Is the Connaught Hospital, University of Sierra Leone Teach Hospital complex chosen for the study because of HIV prevalence in Freetown or/and have the largest number of PLHIV patients in the country? What is the percentage of PLHIV in Sierra Leone treated in this hospital? Can the results found among PLHIV at the Connaught Hospital be generalized to other hospital in Sierra Leone?

Line 94, the authors mentioned that more than 5000 PLHIV are enrolled in ART, with approximately 10 new patients initiating ART per week in the Connaught Hospital. However, there were only 103 PLHIV were enrolled in the study.  Was there a target sample size?  What might be the reasons that there are only 103 PLHIV (2%) enrolled in the study?

Line 134, Were the distributions of socio-demographic and clinical features of these 103 study participants similar to the other PLHIV who did not participate in this study? I am wondering if the finding from these 103 participants be generalized to the other 4900 PLHIV who did not participate in this study?

Line 161, Please consider using “proportion” to replace “prevalence”. Based on the section 2.3, the study participants are convenient samples. It is likely that the PLHIV with eye problem are more likely to participate in this study (selection bias). There is also no information to know if the 103 PLHIV can represent the other 4900 PLHIV at this hospital.

Table 1, Please check the precents (%). Some of the categories do not sum up to 1.  For example, unemployment status, opportunistic infections, etc.

Table 4, the second column, consider presenting raw percentages which align better with univariate and multivariable analysis.

The authors conducted a cross-sectional study to determine 1) the prevalence and types of ophthalmic abnormalities in PLHIV, 2) the risk factors for ophthalmic abnormalities among PLHIV at the HIV and eye clinics of Connaught Hospital, University of Sierra Leone Teaching Hospital complex in Freetown, Sierra Leone. It will be helpful that the authors can clarify whether the study participants can represent the PLHIV at the Connaught Hospital and the findings can be linked or generalized to the PLHIV in the country.

Line 73-81, The authors mentioned that this is the first study in Sierra Leone to determine the pattern of ophthalmic manifestations of PLHIV and to provide the necessary evidence to the policy makers for training health care workers and initiating routine eye examinations. Is the Connaught Hospital, University of Sierra Leone Teach Hospital complex chosen for the study because of HIV prevalence in Freetown or/and have the largest number of PLHIV patients in the country? What is the percentage of PLHIV in Sierra Leone treated in this hospital? Can the results found among PLHIV at the Connaught Hospital be generalized to other hospital in Sierra Leone?

Line 94, the authors mentioned that more than 5000 PLHIV are enrolled in ART, with approximately 10 new patients initiating ART per week in the Connaught Hospital. However, there were only 103 PLHIV were enrolled in the study.  Was there a target sample size?  What might be the reasons that there are only 103 PLHIV (2%) enrolled in the study?

Line 134, Were the distributions of socio-demographic and clinical features of these 103 study participants similar to the other PLHIV who did not participate in this study? I am wondering if the finding from these 103 participants be generalized to the other 4900 PLHIV who did not participate in this study?

Line 161, Please consider using “proportion” to replace “prevalence”. Based on the section 2.3, the study participants are convenient samples. It is likely that the PLHIV with eye problem are more likely to participate in this study (selection bias). There is also no information to know if the 103 PLHIV can represent the other 4900 PLHIV at this hospital.

Table 1, Please check the precents (%). Some of the categories do not sum up to 1.  For example, unemployment status, opportunistic infections, etc.

Table 4, the second column, consider presenting raw percentages which align better with univariate and multivariable analysis.

The authors conducted a cross-sectional study to determine 1) the prevalence and types of ophthalmic abnormalities in PLHIV, 2) the risk factors for ophthalmic abnormalities among PLHIV at the HIV and eye clinics of Connaught Hospital, University of Sierra Leone Teaching Hospital complex in Freetown, Sierra Leone. It will be helpful that the authors can clarify whether the study participants can represent the PLHIV at the Connaught Hospital and the findings can be linked or generalized to the PLHIV in the country.

Line 73-81, The authors mentioned that this is the first study in Sierra Leone to determine the pattern of ophthalmic manifestations of PLHIV and to provide the necessary evidence to the policy makers for training health care workers and initiating routine eye examinations. Is the Connaught Hospital, University of Sierra Leone Teach Hospital complex chosen for the study because of HIV prevalence in Freetown or/and have the largest number of PLHIV patients in the country? What is the percentage of PLHIV in Sierra Leone treated in this hospital? Can the results found among PLHIV at the Connaught Hospital be generalized to other hospital in Sierra Leone?

Line 94, the authors mentioned that more than 5000 PLHIV are enrolled in ART, with approximately 10 new patients initiating ART per week in the Connaught Hospital. However, there were only 103 PLHIV were enrolled in the study.  Was there a target sample size?  What might be the reasons that there are only 103 PLHIV (2%) enrolled in the study?

Line 134, Were the distributions of socio-demographic and clinical features of these 103 study participants similar to the other PLHIV who did not participate in this study? I am wondering if the finding from these 103 participants be generalized to the other 4900 PLHIV who did not participate in this study?

Line 161, Please consider using “proportion” to replace “prevalence”. Based on the section 2.3, the study participants are convenient samples. It is likely that the PLHIV with eye problem are more likely to participate in this study (selection bias). There is also no information to know if the 103 PLHIV can represent the other 4900 PLHIV at this hospital.

Table 1, Please check the precents (%). Some of the categories do not sum up to 1.  For example, unemployment status, opportunistic infections, etc.

Table 4, the second column, consider presenting raw percentages which align better with univariate and multivariable analysis.

The authors conducted a cross-sectional study to determine 1) the prevalence and types of ophthalmic abnormalities in PLHIV, 2) the risk factors for ophthalmic abnormalities among PLHIV at the HIV and eye clinics of Connaught Hospital, University of Sierra Leone Teaching Hospital complex in Freetown, Sierra Leone. It will be helpful that the authors can clarify whether the study participants can represent the PLHIV at the Connaught Hospital and the findings can be linked or generalized to the PLHIV in the country.

Line 73-81, The authors mentioned that this is the first study in Sierra Leone to determine the pattern of ophthalmic manifestations of PLHIV and to provide the necessary evidence to the policy makers for training health care workers and initiating routine eye examinations. Is the Connaught Hospital, University of Sierra Leone Teach Hospital complex chosen for the study because of HIV prevalence in Freetown or/and have the largest number of PLHIV patients in the country? What is the percentage of PLHIV in Sierra Leone treated in this hospital? Can the results found among PLHIV at the Connaught Hospital be generalized to other hospital in Sierra Leone?

Line 94, the authors mentioned that more than 5000 PLHIV are enrolled in ART, with approximately 10 new patients initiating ART per week in the Connaught Hospital. However, there were only 103 PLHIV were enrolled in the study.  Was there a target sample size?  What might be the reasons that there are only 103 PLHIV (2%) enrolled in the study?

Line 134, Were the distributions of socio-demographic and clinical features of these 103 study participants similar to the other PLHIV who did not participate in this study? I am wondering if the finding from these 103 participants be generalized to the other 4900 PLHIV who did not participate in this study?

Line 161, Please consider using “proportion” to replace “prevalence”. Based on the section 2.3, the study participants are convenient samples. It is likely that the PLHIV with eye problem are more likely to participate in this study (selection bias). There is also no information to know if the 103 PLHIV can represent the other 4900 PLHIV at this hospital.

Table 1, Please check the precents (%). Some of the categories do not sum up to 1.  For example, unemployment status, opportunistic infections, etc.

Table 4, the second column, consider presenting raw percentages which align better with univariate and multivariable analysis.

The authors conducted a cross-sectional study to determine 1) the prevalence and types of ophthalmic abnormalities in PLHIV, 2) the risk factors for ophthalmic abnormalities among PLHIV at the HIV and eye clinics of Connaught Hospital, University of Sierra Leone Teaching Hospital complex in Freetown, Sierra Leone. It will be helpful that the authors can clarify whether the study participants can represent the PLHIV at the Connaught Hospital and the findings can be linked or generalized to the PLHIV in the country.

Line 73-81, The authors mentioned that this is the first study in Sierra Leone to determine the pattern of ophthalmic manifestations of PLHIV and to provide the necessary evidence to the policy makers for training health care workers and initiating routine eye examinations. Is the Connaught Hospital, University of Sierra Leone Teach Hospital complex chosen for the study because of HIV prevalence in Freetown or/and have the largest number of PLHIV patients in the country? What is the percentage of PLHIV in Sierra Leone treated in this hospital? Can the results found among PLHIV at the Connaught Hospital be generalized to other hospital in Sierra Leone?

Line 94, the authors mentioned that more than 5000 PLHIV are enrolled in ART, with approximately 10 new patients initiating ART per week in the Connaught Hospital. However, there were only 103 PLHIV were enrolled in the study.  Was there a target sample size?  What might be the reasons that there are only 103 PLHIV (2%) enrolled in the study?

Line 134, Were the distributions of socio-demographic and clinical features of these 103 study participants similar to the other PLHIV who did not participate in this study? I am wondering if the finding from these 103 participants be generalized to the other 4900 PLHIV who did not participate in this study?

Line 161, Please consider using “proportion” to replace “prevalence”. Based on the section 2.3, the study participants are convenient samples. It is likely that the PLHIV with eye problem are more likely to participate in this study (selection bias). There is also no information to know if the 103 PLHIV can represent the other 4900 PLHIV at this hospital.

Table 1, Please check the precents (%). Some of the categories do not sum up to 1.  For example, unemployment status, opportunistic infections, etc.

Table 4, the second column, consider presenting raw percentages which align better with univariate and multivariable analysis.

The authors conducted a cross-sectional study to determine 1) the prevalence and types of ophthalmic abnormalities in PLHIV, 2) the risk factors for ophthalmic abnormalities among PLHIV at the HIV and eye clinics of Connaught Hospital, University of Sierra Leone Teaching Hospital complex in Freetown, Sierra Leone. It will be helpful that the authors can clarify whether the study participants can represent the PLHIV at the Connaught Hospital and the findings can be linked or generalized to the PLHIV in the country.

Line 73-81, The authors mentioned that this is the first study in Sierra Leone to determine the pattern of ophthalmic manifestations of PLHIV and to provide the necessary evidence to the policy makers for training health care workers and initiating routine eye examinations. Is the Connaught Hospital, University of Sierra Leone Teach Hospital complex chosen for the study because of HIV prevalence in Freetown or/and have the largest number of PLHIV patients in the country? What is the percentage of PLHIV in Sierra Leone treated in this hospital? Can the results found among PLHIV at the Connaught Hospital be generalized to other hospital in Sierra Leone?

Line 94, the authors mentioned that more than 5000 PLHIV are enrolled in ART, with approximately 10 new patients initiating ART per week in the Connaught Hospital. However, there were only 103 PLHIV were enrolled in the study.  Was there a target sample size?  What might be the reasons that there are only 103 PLHIV (2%) enrolled in the study?

Line 134, Were the distributions of socio-demographic and clinical features of these 103 study participants similar to the other PLHIV who did not participate in this study? I am wondering if the finding from these 103 participants be generalized to the other 4900 PLHIV who did not participate in this study?

Line 161, Please consider using “proportion” to replace “prevalence”. Based on the section 2.3, the study participants are convenient samples. It is likely that the PLHIV with eye problem are more likely to participate in this study (selection bias). There is also no information to know if the 103 PLHIV can represent the other 4900 PLHIV at this hospital.

Table 1, Please check the precents (%). Some of the categories do not sum up to 1.  For example, unemployment status, opportunistic infections, etc.

Table 4, the second column, consider presenting raw percentages which align better with univariate and multivariable analysis.

The authors conducted a cross-sectional study to determine 1) the prevalence and types of ophthalmic abnormalities in PLHIV, 2) the risk factors for ophthalmic abnormalities among PLHIV at the HIV and eye clinics of Connaught Hospital, University of Sierra Leone Teaching Hospital complex in Freetown, Sierra Leone. It will be helpful that the authors can clarify whether the study participants can represent the PLHIV at the Connaught Hospital and the findings can be linked or generalized to the PLHIV in the country.

Line 73-81, The authors mentioned that this is the first study in Sierra Leone to determine the pattern of ophthalmic manifestations of PLHIV and to provide the necessary evidence to the policy makers for training health care workers and initiating routine eye examinations. Is the Connaught Hospital, University of Sierra Leone Teach Hospital complex chosen for the study because of HIV prevalence in Freetown or/and have the largest number of PLHIV patients in the country? What is the percentage of PLHIV in Sierra Leone treated in this hospital? Can the results found among PLHIV at the Connaught Hospital be generalized to other hospital in Sierra Leone?

Line 94, the authors mentioned that more than 5000 PLHIV are enrolled in ART, with approximately 10 new patients initiating ART per week in the Connaught Hospital. However, there were only 103 PLHIV were enrolled in the study.  Was there a target sample size?  What might be the reasons that there are only 103 PLHIV (2%) enrolled in the study?

Line 134, Were the distributions of socio-demographic and clinical features of these 103 study participants similar to the other PLHIV who did not participate in this study? I am wondering if the finding from these 103 participants be generalized to the other 4900 PLHIV who did not participate in this study?

Line 161, Please consider using “proportion” to replace “prevalence”. Based on the section 2.3, the study participants are convenient samples. It is likely that the PLHIV with eye problem are more likely to participate in this study (selection bias). There is also no information to know if the 103 PLHIV can represent the other 4900 PLHIV at this hospital.

Table 1, Please check the percents (%). Some of the categories do not sum up to 1.  For example, unemployment status, opportunistic infections, etc.

Table 4, the second column, consider presenting raw percentages which align better with univariate and multivariable analysis.

The authors conducted a cross-sectional study to determine 1) the prevalence and types of ophthalmic abnormalities in PLHIV, 2) the risk factors for ophthalmic abnormalities among PLHIV at the HIV and eye clinics of Connaught Hospital, University of Sierra Leone Teaching Hospital complex in Freetown, Sierra Leone. It will be helpful that the authors can clarify whether the study participants can represent the PLHIV at the Connaught Hospital and the findings can be linked or generalized to the PLHIV in the country.

Line 73-81, The authors mentioned that this is the first study in Sierra Leone to determine the pattern of ophthalmic manifestations of PLHIV and to provide the necessary evidence to the policy makers for training health care workers and initiating routine eye examinations. Is the Connaught Hospital, University of Sierra Leone Teach Hospital complex chosen for the study because of HIV prevalence in Freetown or/and have the largest number of PLHIV patients in the country? What is the percentage of PLHIV in Sierra Leone treated in this hospital? Can the results found among PLHIV at the Connaught Hospital be generalized to other hospital in Sierra Leone?

Line 94, the authors mentioned that more than 5000 PLHIV are enrolled in ART, with approximately 10 new patients initiating ART per week in the Connaught Hospital. However, there were only 103 PLHIV were enrolled in the study.  Was there a target sample size?  What might be the reasons that there are only 103 PLHIV (2%) enrolled in the study?

Line 134, Were the distributions of socio-demographic and clinical features of these 103 study participants similar to the other PLHIV who did not participate in this study? I am wondering if the finding from these 103 participants be generalized to the other 4900 PLHIV who did not participate in this study?

Line 161, Please consider using “proportion” to replace “prevalence”. Based on the section 2.3, the study participants are convenient samples. It is likely that the PLHIV with eye problem are more likely to participate in this study (selection bias). There is also no information to know if the 103 PLHIV can represent the other 4900 PLHIV at this hospital.

Table 1, Please check the precents (%). Some of the categories do not sum up to 1.  For example, unemployment status, opportunistic infections, etc.

Table 4, the second column, consider presenting raw percentages which align better with univariate and multivariable analysis.

Author Response

Reviewer 1

1.     The authors conducted a cross-sectional study to determine 1) the prevalence and types of ophthalmic abnormalities in PLHIV, 2) the risk factors for ophthalmic abnormalities among PLHIV at the HIV and eye clinics of Connaught Hospital, University of Sierra Leone Teaching Hospital complex in Freetown, Sierra Leone. It will be helpful that the authors can clarify whether the study participants can represent the PLHIV at the Connaught Hospital and the findings can be linked or generalized to the PLHIV in the country.

Thank you for your observation. It is clear that our findings are not generalizable to the population of HIV in Sierra Leone as the study was conducted in a single center and has a small sample size. This limitation has been highlighted in the manuscript (Lines 239 to 241).

2.     Line 73-81, The authors mentioned that this is the first study in Sierra Leone to determine the pattern of ophthalmic manifestations of PLHIV and to provide the necessary evidence to the policy makers for training health care workers and initiating routine eye examinations. Is the Connaught Hospital, University of Sierra Leone Teach Hospital complex chosen for the study because of HIV prevalence in Freetown or/and have the largest number of PLHIV patients in the country? What is the percentage of PLHIV in Sierra Leone treated in this hospital? Can the results found among PLHIV at the Connaught Hospital be generalized to other hospital in Sierra Leone?

Connaught hospital was chosen for this study because it is the only hospital in Sierra Leone to offer structured eye services and has a large HIV population. We agree that the findings of our study are not generalizable to the PLHIV in Sierra Leone. We stress that this as a limitation.

3.     Line 94, the authors mentioned that more than 5000 PLHIV are enrolled in ART, with approximately 10 new patients initiating ART per week in the Connaught Hospital. However, there were only 103 PLHIV were enrolled in the study.  Was there a target sample size?  What might be the reasons that there are only 103 PLHIV (2%) enrolled in the study?

Thank you for this observation. The sample size was calculated prior to the conduct of the study. We are sorry for omitting information related to this in our initial manuscript submission. We have now added information on the sample size calculation in the paper.  (Lines 99 to 106).

4.     Line 134, Were the distributions of socio-demographic and clinical features of these 103 study participants similar to the other PLHIV who did not participate in this study? I am wondering if the finding from these 103 participants be generalized to the other 4900 PLHIV who did not participate in this study?

As stated earlier, the study was conducted in a single study site and included a relatively small sample size. Thus, the findings are not generalizable to the total population living with HIV in Sierra Leone. We have added this as a limitation to this study in lines 239 to 241.

5.     Line 161, Please consider using “proportion” to replace “prevalence”. Based on the section 2.3, the study participants are convenient samples. It is likely that the PLHIV with eye problem are more likely to participate in this study (selection bias). There is also no information to know if the 103 PLHIV can represent the other 4900 PLHIV at this hospital.

We agree that our study included a relatively small sample size, but that study participants were not recruited based on their complains and this may have minimized selection bias.

6.     Table 1, Please check the precents (%). Some of the categories do not sum up to 1.  For example, unemployment status, opportunistic infections, etc.

We have added a foot note in Table 1 to indicate that some parameters are not mutually exclusive, and not all patients had an opportunistic infection.

7.     Table 4, the second column, consider presenting raw percentages which align better with univariate and multivariable analysis.

The values in the parenthesis of the second column of Table 4 represents the percentages of the raw data. We feel that presenting the data in another way may confuse the audience.

Reviewer 2 Report

This cross sectional study by Mustapha J et.al investigates ophthalmic complications among PLHIV in Sierra Leone. They recruited a total of 103 PLHIV during January 2020- March 2020 for this study.  They have done a through eye assessment among this 103 population. They have used Snellen’s Visual Acuity chart to determine the visual acuity and a torch, loupes, and slit lamps to examine the ocular adnexal and anterior segments. The tear breakup time (TBUT) method was used to assess the ocular surface, and a TBUT of fewer than 10 seconds was considered indicative of dry eye.

They showed that 44.7% of ART experienced PLHIV in Sierra Leone had an ocular manifestations.

This study is well performed and presented well.

check for typo error and few references for example ref.9

Author Response

Reviewer #2:
1. This cross-sectional study by Mustapha J et.al investigates ophthalmic complications among PLHIV in Sierra Leone. They recruited a total of 103 PLHIV during January 2020- March 2020 for this study.  They have done a through eye assessment among these 103 populations. They have used Snellen’s Visual Acuity chart to determine the visual acuity and a torch, loupes, and slit lamps to examine the ocular adnexal and anterior segments. The tear breakup time (TBUT) method was used to assess the ocular surface, and a TBUT of fewer than 10 seconds was considered indicative of dry eye.

They showed that 44.7% of ART experienced PLHIV in Sierra Leone had an ocular manifestation.

This study is well performed and presented well.

Thank you.

2.check for typo error and few references for example ref.9

We have corrected this (Lines 202 and 205)